# Multi-Task Learning as Multi-Objective Optimization

**Ozan Sener**
Intel Labs

**Vladlen Koltun**
Intel Labs

## Abstract

In multi-task learning, multiple tasks are solved jointly, sharing inductive bias between them. Multi-task learning is inherently a multi-objective problem because different tasks may conflict, necessitating a trade-off. A common compromise is to optimize a proxy objective that minimizes a weighted linear combination of per-task losses. However, this workaround is only valid when the tasks do not compete, which is rarely the case. In this paper, we explicitly cast multi-task learning as multi-objective optimization, with the overall objective of finding a Pareto optimal solution. To this end, we use algorithms developed in the gradient-based multi-objective optimization literature. These algorithms are not directly applicable to large-scale learning problems since they scale poorly with the dimensionality of the gradients and the number of tasks. We therefore propose an upper bound for the multi-objective loss and show that it can be optimized efficiently. We further prove that optimizing this upper bound yields a Pareto optimal solution under realistic assumptions. We apply our method to a variety of multi-task deep learning problems including digit classification, scene understanding (joint semantic segmentation, instance segmentation, and depth estimation), and multi-label classification. Our method produces higher-performing models than recent multi-task learning formulations or per-task training.

## 1 Introduction

One of the most surprising results in statistics is Stein's paradox. Stein (1956) showed that it is better to estimate the means of three or more Gaussian random variables using samples from all of them rather than estimating them separately, even when the Gaussians are independent. Stein's paradox was an early motivation for multi-task learning (MTL) (Caruana, 1997), a learning paradigm in which data from multiple tasks is used with the hope to obtain superior performance over learning each task independently. Potential advantages of MTL go beyond the direct implications of Stein's paradox, since even seemingly unrelated real world tasks have strong dependencies due to the shared processes that give rise to the data. For example, although autonomous driving and object manipulation are seemingly unrelated, the underlying data is governed by the same laws of optics, material properties, and dynamics. This motivates the use of multiple tasks as an inductive bias in learning systems.

A typical MTL system is given a collection of input points and sets of targets for various tasks per point. A common way to set up the inductive bias across tasks is to design a parametrized hypothesis class that shares some parameters across tasks. Typically, these parameters are learned by solving an optimization problem that minimizes a weighted sum of the empirical risk for each task. However, the linear-combination formulation is only sensible when there is a parameter set that is effective across all tasks. In other words, minimization of a weighted sum of empirical risk is only valid if tasks are not competing, which is rarely the case. MTL with conflicting objectives requires modeling of the trade-off between tasks, which is beyond what a linear combination achieves.

An alternative objective for MTL is finding solutions that are not dominated by any others. Such solutions are said to be Pareto optimal. In this paper, we cast the objective of MTL in terms of finding Pareto optimal solutions.

The problem of finding Pareto optimal solutions given multiple criteria is called multi-objective optimization. A variety of algorithms for multi-objective optimization exist. One such approach is the multiple-gradient descent algorithm (MGDA), which uses gradient-based optimization and provably converges to a point on the Pareto set (Désidéri, 2012). MGDA is well-suited for multi-task learning with deep networks. It can use the gradients of each task and solve an optimization problem to decide on an update over the shared parameters. However, there are two technical problems that hinder the applicability of MGDA on a large scale. (i) The underlying optimization problem does not scale gracefully to high-dimensional gradients, which arise naturally in deep networks. (ii) The algorithm requires explicit computation of gradients per task, which results in linear scaling of the number of backward passes and roughly multiplies the training time by the number of tasks.

In this paper, we develop a Frank-Wolfe-based optimizer that scales to high-dimensional problems. Furthermore, we provide an upper bound for the MGDA optimization objective and show that it can be computed via a single backward pass without explicit task-specific gradients, thus making the computational overhead of the method negligible. We prove that using our upper bound yields a Pareto optimal solution under realistic assumptions. The result is an exact algorithm for multi-objective optimization of deep networks with negligible computational overhead.

We empirically evaluate the presented method on three different problems. First, we perform an extensive evaluation on multi-digit classification with MultiMNIST (Sabour et al., 2017). Second, we cast multi-label classification as MTL and conduct experiments with the CelebA dataset (Liu et al., 2015b). Lastly, we apply the presented method to scene understanding; specifically, we perform joint semantic segmentation, instance segmentation, and depth estimation on the Cityscapes dataset (Cordts et al., 2016). The number of tasks in our evaluation varies from 2 to 40. Our method clearly outperforms all baselines.

## 2 Related Work

**Multi-task learning.** We summarize the work most closely related to ours and refer the interested reader to reviews by Ruder (2017) and Zhou et al. (2011b) for additional background. Multi-task learning (MTL) is typically conducted via hard or soft parameter sharing. In hard parameter sharing, a subset of the parameters is shared between tasks while other parameters are task-specific. In soft parameter sharing, all parameters are task-specific but they are jointly constrained via Bayesian priors (Xue et al., 2007; Bakker and Heskes, 2003) or a joint dictionary (Argyriou et al., 2007; Long and Wang, 2015; Yang and Hospedales, 2016; Ruder, 2017). We focus on hard parameter sharing with gradient-based optimization, following the success of deep MTL in computer vision (Bilen and Vedaldi, 2016; Misra et al., 2016; Rudd et al., 2016; Yang and Hospedales, 2016; Kokkinos, 2017; Zamir et al., 2018), natural language processing (Collobert and Weston, 2008; Dong et al., 2015; Liu et al., 2015a; Luong et al., 2015; Hashimoto et al., 2017), speech processing (Huang et al., 2013; Seltzer and Droppo, 2013; Huang et al., 2015), and even seemingly unrelated domains over multiple modalities (Kaiser et al., 2017).

Baxter (2000) theoretically analyze the MTL problem as interaction between individual learners and a meta-algorithm. Each learner is responsible for one task and a meta-algorithm decides how the shared parameters are updated. All aforementioned MTL algorithms use weighted summation as the meta-algorithm. Meta-algorithms that go beyond weighted summation have also been explored. Li et al. (2014) consider the case where each individual learner is based on kernel learning and utilize multi-objective optimization. Zhang and Yeung (2010) consider the case where each learner is a linear model and use a task affinity matrix. Zhou et al. (2011a) and Bagherjeiran et al. (2005) use the assumption that tasks share a dictionary and develop an expectation-maximization-like meta-algorithm. de Miranda et al. (2012) and Zhou et al. (2017b) use swarm optimization. None of these methods apply to gradient-based learning of high-capacity models such as modern deep networks. Kendall et al. (2018) and Chen et al. (2018) propose heuristics based on uncertainty and gradient magnitudes, respectively, and apply their methods to convolutional neural networks. Another recent work uses multi-agent reinforcement learning (Rosenbaum et al., 2017).

**Multi-objective optimization.** Multi-objective optimization addresses the problem of optimizing a set of possibly contrasting objectives. We recommend Miettinen (1998) and Ehrgott (2005) for surveys of this field. Of particular relevance to our work is gradient-based multi-objective optimization, as developed by Fliege and Svaiter (2000), Schäffler et al. (2002), and Désidéri (2012). These methods

use multi-objective Karush-Kuhn-Tucker (KKT) conditions (Kuhn and Tucker, 1951) and find a descent direction that decreases all objectives. This approach was extended to stochastic gradient descent by Peitz and Dellnitz (2018) and Poirion et al. (2017). In machine learning, these methods have been applied to multi-agent learning (Ghosh et al., 2013; Pirotta and Restelli, 2016; Parisi et al., 2014), kernel learning (Li et al., 2014), sequential decision making (Roijers et al., 2013), and Bayesian optimization (Shah and Ghahramani, 2016; Hernández-Lobato et al., 2016). Our work applies gradient-based multi-objective optimization to multi-task learning.

## 3  Multi-Task Learning as Multi-Objective Optimization

Consider a multi-task learning (MTL) problem over an input space $\mathcal{X}$ and a collection of task spaces $\{\mathcal{Y}^t\}_{t \in [T]}$, such that a large dataset of i.i.d. data points $\{\mathbf{x}_i, y_i^1, \ldots, y_i^T\}_{i \in [N]}$ is given where $T$ is the number of tasks, $N$ is the number of data points, and $y_i^t$ is the label of the $t^{\text{th}}$ task for the $i^{\text{th}}$ data point.[1] We further consider a parametric hypothesis class per task as $f^t(\mathbf{x}; \boldsymbol{\theta}^{sh}, \boldsymbol{\theta}^t) : \mathcal{X} \to \mathcal{Y}^t$, such that some parameters ($\boldsymbol{\theta}^{sh}$) are shared between tasks and some ($\boldsymbol{\theta}^t$) are task-specific. We also consider task-specific loss functions $\mathcal{L}^t(\cdot, \cdot) : \mathcal{Y}^t \times \mathcal{Y}^t \to \mathbb{R}^+$.

Although many hypothesis classes and loss functions have been proposed in the MTL literature, they generally yield the following empirical risk minimization formulation:

$$\min_{\substack{\boldsymbol{\theta}^{sh}, \\ \boldsymbol{\theta}^1, \ldots, \boldsymbol{\theta}^T}} \quad \sum_{t=1}^{T} c^t \hat{\mathcal{L}}^t(\boldsymbol{\theta}^{sh}, \boldsymbol{\theta}^t) \tag{1}$$

for some static or dynamically computed weights $c^t$ per task, where $\hat{\mathcal{L}}^t(\boldsymbol{\theta}^{sh}, \boldsymbol{\theta}^t)$ is the empirical loss of the task $t$, defined as $\hat{\mathcal{L}}^t(\boldsymbol{\theta}^{sh}, \boldsymbol{\theta}^t) \triangleq \frac{1}{N} \sum_i \mathcal{L}\big(f^t(\mathbf{x}_i; \boldsymbol{\theta}^{sh}, \boldsymbol{\theta}^t), y_i^t\big)$.

Although the weighted summation formulation (1) is intuitively appealing, it typically either requires an expensive grid search over various scalings or the use of a heuristic (Kendall et al., 2018; Chen et al., 2018). A basic justification for scaling is that it is not possible to define global optimality in the MTL setting. Consider two sets of solutions $\boldsymbol{\theta}$ and $\bar{\boldsymbol{\theta}}$ such that $\hat{\mathcal{L}}^{t_1}(\boldsymbol{\theta}^{sh}, \boldsymbol{\theta}^{t_1}) < \hat{\mathcal{L}}^{t_1}(\bar{\boldsymbol{\theta}}^{sh}, \bar{\boldsymbol{\theta}}^{t_1})$ and $\hat{\mathcal{L}}^{t_2}(\boldsymbol{\theta}^{sh}, \boldsymbol{\theta}^{t_2}) > \hat{\mathcal{L}}^{t_2}(\bar{\boldsymbol{\theta}}^{sh}, \bar{\boldsymbol{\theta}}^{t_2})$, for some tasks $t_1$ and $t_2$. In other words, solution $\boldsymbol{\theta}$ is better for task $t_1$ whereas $\bar{\boldsymbol{\theta}}$ is better for $t_2$. It is not possible to compare these two solutions without a pairwise importance of tasks, which is typically not available.

Alternatively, MTL can be formulated as multi-objective optimization: optimizing a collection of possibly conflicting objectives. This is the approach we take. We specify the multi-objective optimization formulation of MTL using a vector-valued loss $\mathbf{L}$:

$$\min_{\substack{\boldsymbol{\theta}^{sh}, \\ \boldsymbol{\theta}^1, \ldots, \boldsymbol{\theta}^T}} \mathbf{L}(\boldsymbol{\theta}^{sh}, \boldsymbol{\theta}^1, \ldots, \boldsymbol{\theta}^T) = \min_{\substack{\boldsymbol{\theta}^{sh}, \\ \boldsymbol{\theta}^1, \ldots, \boldsymbol{\theta}^T}} \big(\hat{\mathcal{L}}^1(\boldsymbol{\theta}^{sh}, \boldsymbol{\theta}^1), \ldots, \hat{\mathcal{L}}^T(\boldsymbol{\theta}^{sh}, \boldsymbol{\theta}^T)\big)^{\mathsf{T}}. \tag{2}$$

The goal of multi-objective optimization is achieving Pareto optimality.

**Definition 1** (Pareto optimality for MTL)

(a) *A solution $\boldsymbol{\theta}$ dominates a solution $\bar{\boldsymbol{\theta}}$ if $\hat{\mathcal{L}}^t(\boldsymbol{\theta}^{sh}, \boldsymbol{\theta}^t) \leq \hat{\mathcal{L}}^t(\bar{\boldsymbol{\theta}}^{sh}, \bar{\boldsymbol{\theta}}^t)$ for all tasks $t$ and $\mathbf{L}(\boldsymbol{\theta}^{sh}, \boldsymbol{\theta}^1, \ldots, \boldsymbol{\theta}^T) \neq \mathbf{L}(\bar{\boldsymbol{\theta}}^{sh}, \bar{\boldsymbol{\theta}}^1, \ldots, \bar{\boldsymbol{\theta}}^T)$.*

(b) *A solution $\boldsymbol{\theta}^\star$ is called Pareto optimal if there exists no solution $\boldsymbol{\theta}$ that dominates $\boldsymbol{\theta}^\star$.*

The set of Pareto optimal solutions is called the Pareto set ($\mathcal{P}_{\boldsymbol{\theta}}$) and its image is called the Pareto front ($\mathcal{P}_{\mathbf{L}} = \{\mathbf{L}(\boldsymbol{\theta})\}_{\boldsymbol{\theta} \in \mathcal{P}_{\boldsymbol{\theta}}}$). In this paper, we focus on gradient-based multi-objective optimization due to its direct relevance to gradient-based MTL.

In the rest of this section, we first summarize in Section 3.1 how multi-objective optimization can be performed with gradient descent. Then, we suggest in Section 3.2 a practical algorithm for performing multi-objective optimization over very large parameter spaces. Finally, in Section 3.3 we propose an efficient solution for multi-objective optimization designed directly for high-capacity deep networks. Our method scales to very large models and a high number of tasks with negligible overhead.

## 3.1 Multiple Gradient Descent Algorithm

As in the single-objective case, multi-objective optimization can be solved to local optimality via gradient descent. In this section, we summarize one such approach, called the multiple gradient descent algorithm (MGDA) (Désidéri, 2012). MGDA leverages the Karush-Kuhn-Tucker (KKT) conditions, which are necessary for optimality (Fliege and Svaiter, 2000; Schäffler et al., 2002; Désidéri, 2012). We now state the KKT conditions for both task-specific and shared parameters:

- There exist $\alpha^1, \ldots, \alpha^T \geq 0$ such that $\sum_{t=1}^{T} \alpha^t = 1$ and $\sum_{t=1}^{T} \alpha^t \nabla_{\boldsymbol{\theta}^{sh}} \hat{\mathcal{L}}^t(\boldsymbol{\theta}^{sh}, \boldsymbol{\theta}^t) = 0$

- For all tasks $t$, $\nabla_{\boldsymbol{\theta}^t} \hat{\mathcal{L}}^t(\boldsymbol{\theta}^{sh}, \boldsymbol{\theta}^t) = 0$

Any solution that satisfies these conditions is called a Pareto stationary point. Although every Pareto optimal point is Pareto stationary, the reverse may not be true. Consider the optimization problem

$$\min_{\alpha^1, \ldots, \alpha^T} \left\{ \left\| \sum_{t=1}^{T} \alpha^t \nabla_{\boldsymbol{\theta}^{sh}} \hat{\mathcal{L}}^t(\boldsymbol{\theta}^{sh}, \boldsymbol{\theta}^t) \right\|_2^2 \; \middle| \; \sum_{t=1}^{T} \alpha^t = 1, \alpha^t \geq 0 \quad \forall t \right\} \tag{3}$$

Désidéri (2012) showed that either the solution to this optimization problem is $0$ and the resulting point satisfies the KKT conditions, or the solution gives a descent direction that improves all tasks. Hence, the resulting MTL algorithm would be gradient descent on the task-specific parameters followed by solving (3) and applying the solution ($\sum_{t=1}^{T} \alpha^t \nabla_{\boldsymbol{\theta}^{sh}}$) as a gradient update to shared parameters. We discuss how to solve (3) for an arbitrary model in Section 3.2 and present an efficient solution when the underlying model is an encoder-decoder in Section 3.3.

## 3.2 Solving the Optimization Problem

The optimization problem defined in (3) is equivalent to finding a minimum-norm point in the convex hull of the set of input points. This problem arises naturally in computational geometry: it is equivalent to finding the closest point within a convex hull to a given query point. It has been studied extensively (Makimoto et al., 1994; Wolfe, 1976; Sekitani and Yamamoto, 1993). Although many algorithms have been proposed, they do not apply in our setting because the assumptions they make do not hold. Algorithms proposed in the computational geometry literature address the problem of finding minimum-norm points in the convex hull of a large number of points in a low-dimensional space (typically of dimensionality 2 or 3). In our setting, the number of points is the number of tasks and is typically low; in contrast, the dimensionality is the number of shared parameters and can be in the millions. We therefore use a different approach based on convex optimization, since (3) is a convex quadratic problem with linear constraints.

Before we tackle the general case, let's consider the case of two tasks. The optimization problem can be defined as $\min_{\alpha \in [0,1]} \|\alpha \nabla_{\boldsymbol{\theta}^{sh}} \hat{\mathcal{L}}^1(\boldsymbol{\theta}^{sh}, \boldsymbol{\theta}^1) + (1-\alpha) \nabla_{\boldsymbol{\theta}^{sh}} \hat{\mathcal{L}}^2(\boldsymbol{\theta}^{sh}, \boldsymbol{\theta}^2)\|_2^2$, which is a one-dimensional quadratic function of $\alpha$ with an analytical solution:

$$\hat{\alpha} = \left[ \frac{\left( \nabla_{\boldsymbol{\theta}^{sh}} \hat{\mathcal{L}}^2(\boldsymbol{\theta}^{sh}, \boldsymbol{\theta}^2) - \nabla_{\boldsymbol{\theta}^{sh}} \hat{\mathcal{L}}^1(\boldsymbol{\theta}^{sh}, \boldsymbol{\theta}^1) \right)^{\mathsf{T}} \nabla_{\boldsymbol{\theta}^{sh}} \hat{\mathcal{L}}^2(\boldsymbol{\theta}^{sh}, \boldsymbol{\theta}^2)}{\|\nabla_{\boldsymbol{\theta}^{sh}} \hat{\mathcal{L}}^1(\boldsymbol{\theta}^{sh}, \boldsymbol{\theta}^1) - \nabla_{\boldsymbol{\theta}^{sh}} \hat{\mathcal{L}}^2(\boldsymbol{\theta}^{sh}, \boldsymbol{\theta}^2)\|_2^2} \right]_{+, \frac{1}{\mathsf{T}}} \tag{4}$$

where $[\cdot]_{+, \frac{1}{\mathsf{T}}}$ represents clipping to $[0, 1]$ as $[a]_{+, \frac{1}{\mathsf{T}}} = \max(\min(a, 1), 0)$. We further visualize this solution in Figure 1. Although this is only applicable when $T = 2$, this enables efficient application of the Frank-Wolfe algorithm (Jaggi, 2013) since the line search can be solved analytically. Hence, we use Frank-Wolfe to solve the constrained optimization problem, using (4) as a subroutine for the line search. We give all the update equations for the Frank-Wolfe solver in Algorithm 2.

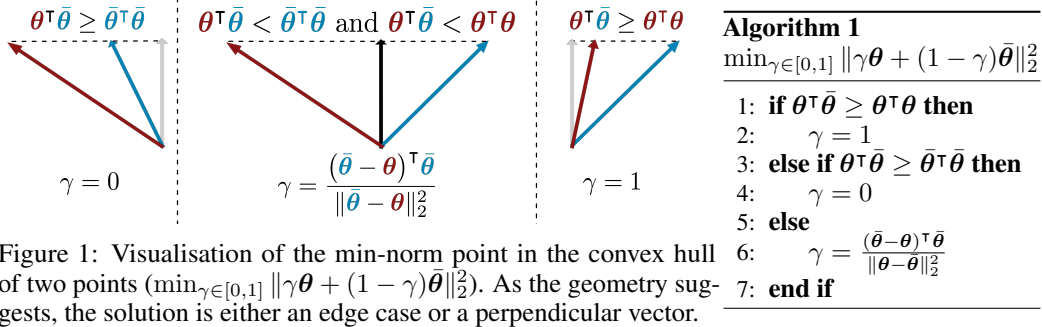

Figure 1: Visualisation of the min-norm point in the convex hull of two points $(\min_{\gamma \in [0,1]} \|\gamma \boldsymbol{\theta} + (1-\gamma)\bar{\boldsymbol{\theta}}\|_2^2)$. As the geometry suggests, the solution is either an edge case or a perpendicular vector.

---

**Algorithm 1**
$\min_{\gamma \in [0,1]} \|\gamma \boldsymbol{\theta} + (1-\gamma)\bar{\boldsymbol{\theta}}\|_2^2$

1: **if** $\boldsymbol{\theta}^\mathsf{T}\bar{\boldsymbol{\theta}} \geq \boldsymbol{\theta}^\mathsf{T}\boldsymbol{\theta}$ **then**
2: $\quad \gamma = 1$
3: **else if** $\boldsymbol{\theta}^\mathsf{T}\bar{\boldsymbol{\theta}} \geq \bar{\boldsymbol{\theta}}^\mathsf{T}\bar{\boldsymbol{\theta}}$ **then**
4: $\quad \gamma = 0$
5: **else**
6: $\quad \gamma = \frac{(\bar{\boldsymbol{\theta}} - \boldsymbol{\theta})^\mathsf{T}\bar{\boldsymbol{\theta}}}{\|\boldsymbol{\theta} - \bar{\boldsymbol{\theta}}\|_2^2}$
7: **end if**

---

**Algorithm 2** Update Equations for MTL

1: **for** $t = 1$ **to** $T$ **do**
2: $\quad \boldsymbol{\theta}^t = \boldsymbol{\theta}^t - \eta \nabla_{\boldsymbol{\theta}^t} \hat{\mathcal{L}}^t(\boldsymbol{\theta}^{sh}, \boldsymbol{\theta}^t)$ $\qquad\qquad\qquad$ ▷ Gradient descent on task-specific parameters
3: **end for**
4: $\alpha^1, \ldots, \alpha^T = \text{FRANKWOLFESOLVER}(\boldsymbol{\theta})$ $\qquad\quad$ ▷ Solve (3) to find a common descent direction
5: $\boldsymbol{\theta}^{sh} = \boldsymbol{\theta}^{sh} - \eta \sum_{t=1}^{T} \alpha^t \nabla_{\boldsymbol{\theta}^{sh}} \hat{\mathcal{L}}^t(\boldsymbol{\theta}^{sh}, \boldsymbol{\theta}^t)$ $\qquad\qquad$ ▷ Gradient descent on shared parameters

6: **procedure** FRANKWOLFESOLVER($\boldsymbol{\theta}$)
7: $\quad$ Initialize $\boldsymbol{\alpha} = (\alpha^1, \ldots, \alpha^T) = (\frac{1}{T}, \ldots, \frac{1}{T})$
8: $\quad$ Precompute $\mathbf{M}$ st. $\mathbf{M}_{i,j} = \left(\nabla_{\boldsymbol{\theta}^{sh}} \hat{\mathcal{L}}^i(\boldsymbol{\theta}^{sh}, \boldsymbol{\theta}^i)\right)^\mathsf{T} \left(\nabla_{\boldsymbol{\theta}^{sh}} \hat{\mathcal{L}}^j(\boldsymbol{\theta}^{sh}, \boldsymbol{\theta}^j)\right)$
9: $\quad$ **repeat**
10: $\qquad \hat{t} = \arg\min_r \sum_t \alpha^t \mathbf{M}_{rt}$
11: $\qquad \hat{\gamma} = \arg\min_\gamma \left((1-\gamma)\boldsymbol{\alpha} + \gamma \boldsymbol{e}_{\hat{t}}\right)^\mathsf{T} \mathbf{M} \left((1-\gamma)\boldsymbol{\alpha} + \gamma \boldsymbol{e}_{\hat{t}}\right)$ $\qquad$ ▷ Using Algorithm 1
12: $\qquad \boldsymbol{\alpha} = (1-\hat{\gamma})\boldsymbol{\alpha} + \hat{\gamma}\boldsymbol{e}_{\hat{t}}$
13: $\quad$ **until** $\hat{\gamma} \sim 0$ **or** Number of Iterations Limit
14: $\quad$ **return** $\alpha^1, \ldots, \alpha^T$
15: **end procedure**

---

### 3.3 Efficient Optimization for Encoder-Decoder Architectures

The MTL update described in Algorithm 2 is applicable to any problem that uses optimization based on gradient descent. Our experiments also suggest that the Frank-Wolfe solver is efficient and accurate as it typically converges in a modest number of iterations with negligible effect on training time. However, the algorithm we described needs to compute $\nabla_{\boldsymbol{\theta}^{sh}} \hat{\mathcal{L}}^t(\boldsymbol{\theta}^{sh}, \boldsymbol{\theta}^t)$ for each task $t$, which requires a backward pass over the shared parameters for each task. Hence, the resulting gradient computation would be the forward pass followed by $T$ backward passes. Considering the fact that computation of the backward pass is typically more expensive than the forward pass, this results in linear scaling of the training time and can be prohibitive for problems with more than a few tasks.

We now propose an efficient method that optimizes an upper bound of the objective and requires only a single backward pass. We further show that optimizing this upper bound yields a Pareto optimal solution under realistic assumptions. The architectures we address conjoin a shared representation function with task-specific decision functions. This class of architectures covers most of the existing deep MTL models and can be formally defined by constraining the hypothesis class as

$$f^t(\mathbf{x}; \boldsymbol{\theta}^{sh}, \boldsymbol{\theta}^t) = (f^t(\cdot; \boldsymbol{\theta}^t) \circ g(\cdot; \boldsymbol{\theta}^{sh}))(\mathbf{x}) = f^t(g(\mathbf{x}; \boldsymbol{\theta}^{sh}); \boldsymbol{\theta}^t) \qquad (5)$$

where $g$ is the representation function shared by all tasks and $f^t$ are the task-specific functions that take this representation as input. If we denote the representations as $\mathbf{Z} = (\mathbf{z}_1, \ldots, \mathbf{z}_N)$, where $\mathbf{z}_i = g(\mathbf{x}_i; \boldsymbol{\theta}^{sh})$, we can state the following upper bound as a direct consequence of the chain rule:

$$\left\| \sum_{t=1}^{T} \alpha^t \nabla_{\boldsymbol{\theta}^{sh}} \hat{\mathcal{L}}^t(\boldsymbol{\theta}^{sh}, \boldsymbol{\theta}^t) \right\|_2^2 \leq \left\| \frac{\partial \mathbf{Z}}{\partial \boldsymbol{\theta}^{sh}} \right\|_2^2 \left\| \sum_{t=1}^{T} \alpha^t \nabla_{\mathbf{Z}} \hat{\mathcal{L}}^t(\boldsymbol{\theta}^{sh}, \boldsymbol{\theta}^t) \right\|_2^2 \qquad (6)$$

where $\left\| \frac{\partial \mathbf{Z}}{\partial \boldsymbol{\theta}^{sh}} \right\|_2$ is the matrix norm of the Jacobian of $\mathbf{Z}$ with respect to $\boldsymbol{\theta}^{sh}$. Two desirable properties of this upper bound are that (i) $\nabla_{\mathbf{Z}} \hat{\mathcal{L}}^t(\boldsymbol{\theta}^{sh}, \boldsymbol{\theta}^t)$ can be computed in a single backward pass for all

tasks and (ii) $\left\|\frac{\partial \mathbf{Z}}{\partial \boldsymbol{\theta}^{sh}}\right\|_2^2$ is not a function of $\alpha^1, \ldots, \alpha^T$, hence it can be removed when it is used as an optimization objective. We replace the $\left\|\sum_{t=1}^T \alpha^t \nabla_{\boldsymbol{\theta}^{sh}} \hat{\mathcal{L}}^t(\boldsymbol{\theta}^{sh}, \boldsymbol{\theta}^t)\right\|_2^2$ term with the upper bound we have just derived in order to obtain the approximate optimization problem and drop the $\left\|\frac{\partial \mathbf{Z}}{\partial \boldsymbol{\theta}^{sh}}\right\|_2^2$ term since it does not affect the optimization. The resulting optimization problem is

$$\min_{\alpha^1, \ldots, \alpha^T} \left\{ \left\| \sum_{t=1}^T \alpha^t \nabla_{\mathbf{Z}} \hat{\mathcal{L}}^t(\boldsymbol{\theta}^{sh}, \boldsymbol{\theta}^t) \right\|_2^2 \Bigg| \sum_{t=1}^T \alpha^t = 1, \alpha^t \geq 0 \quad \forall t \right\} \quad \text{(MGDA-UB)}$$

We refer to this problem as MGDA-UB (Multiple Gradient Descent Algorithm – Upper Bound). In practice, MGDA-UB corresponds to using the gradients of the task losses with respect to the representations instead of the shared parameters. We use Algorithm 2 with only this change as the final method.

Although MGDA-UB is an approximation of the original optimization problem, we now state a theorem that shows that our method produces a Pareto optimal solution under mild assumptions. The proof is given in the supplement.

**Theorem 1** *Assume $\frac{\partial \mathbf{Z}}{\partial \boldsymbol{\theta}^{sh}}$ is full-rank. If $\alpha^{1,\ldots,T}$ is the solution of MGDA-UB, one of the following is true:*

(a) $\sum_{t=1}^T \alpha^t \nabla_{\boldsymbol{\theta}^{sh}} \hat{\mathcal{L}}^t(\boldsymbol{\theta}^{sh}, \boldsymbol{\theta}^t) = 0$ *and the current parameters are Pareto stationary.*

(b) $\sum_{t=1}^T \alpha^t \nabla_{\boldsymbol{\theta}^{sh}} \hat{\mathcal{L}}^t(\boldsymbol{\theta}^{sh}, \boldsymbol{\theta}^t)$ *is a descent direction that decreases all objectives.*

This result follows from the fact that as long as $\frac{\partial \mathbf{Z}}{\partial \boldsymbol{\theta}^{sh}}$ is full rank, optimizing the upper bound corresponds to minimizing the norm of the convex combination of the gradients using the Mahalonobis norm defined by $\frac{\partial \mathbf{Z}}{\partial \boldsymbol{\theta}^{sh}}^\mathsf{T} \frac{\partial \mathbf{Z}}{\partial \boldsymbol{\theta}^{sh}}$. The non-singularity assumption is reasonable as singularity implies that tasks are linearly related and a trade-off is not necessary. In summary, our method provably finds a Pareto stationary point with negligible computational overhead and can be applied to any deep multi-objective problem with an encoder-decoder model.

## 4 Experiments

We evaluate the presented MTL method on a number of problems. First, we use MultiMNIST (Sabour et al., 2017), an MTL adaptation of MNIST (LeCun et al., 1998). Next, we tackle multi-label classification on the CelebA dataset (Liu et al., 2015b) by considering each label as a distinct binary classification task. These problems include both classification and regression, with the number of tasks ranging from 2 to 40. Finally, we experiment with scene understanding, jointly tackling the tasks of semantic segmentation, instance segmentation, and depth estimation on the Cityscapes dataset (Cordts et al., 2016). We discuss each experiment separately in the following subsections.

The baselines we consider are (i) **uniform scaling:** minimizing a uniformly weighted sum of loss functions $\frac{1}{T} \sum_t \mathcal{L}^t$, (ii) **single task:** solving tasks independently, (iii) **grid search:** exhaustively trying various values from $\{c^t \in [0,1] | \sum_t c^t = 1\}$ and optimizing for $\frac{1}{T} \sum_t c^t \mathcal{L}^t$, (iv) **Kendall et al. (2018):** using the uncertainty weighting proposed by Kendall et al. (2018), and (v) **GradNorm:** using the normalization proposed by Chen et al. (2018).

### 4.1 MultiMNIST

Our initial experiments are on MultiMNIST, an MTL version of the MNIST dataset (Sabour et al., 2017). In order to convert digit classification into a multi-task problem, Sabour et al. (2017) overlaid multiple images together. We use a similar construction. For each image, a different one is chosen uniformly in random. Then one of these images is put at the top-left and the other one is at the bottom-right. The resulting tasks are: classifying the digit on the top-left (task-L) and classifying the digit on the bottom-right (task-R). We use 60K examples and directly apply existing single-task MNIST models. The MultiMNIST dataset is illustrated in the supplement.

We use the LeNet architecture (LeCun et al., 1998). We treat all layers except the last as the representation function $g$ and put two fully-connected layers as task-specific functions (see the

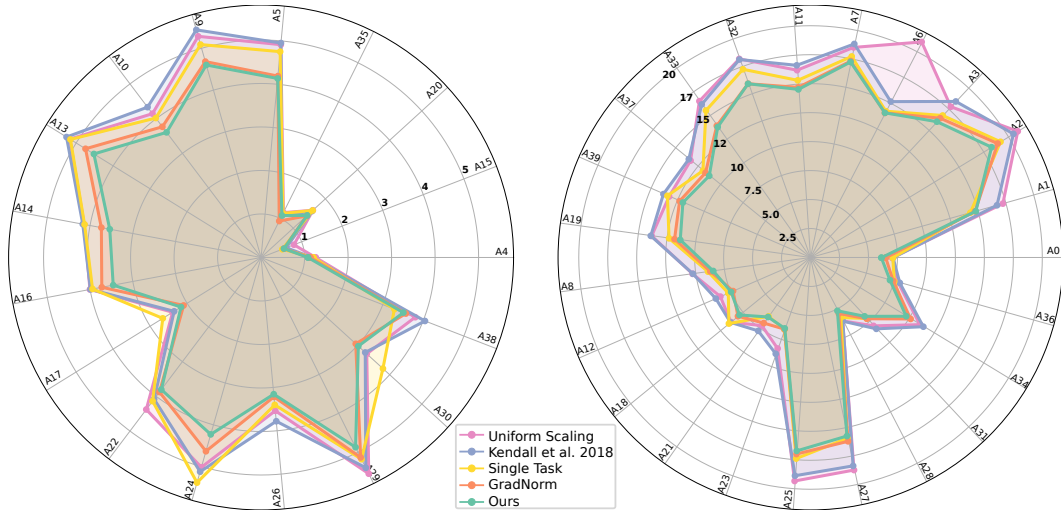

Figure 2: Radar charts of percentage error per attribute on CelebA (Liu et al., 2015b). Lower is better. We divide attributes into two sets for legibility: easy on the left, hard on the right. Zoom in for details.

supplement for details). We visualize the performance profile as a scatter plot of accuracies on task-L and task-R in Figure 3, and list the results in Table 3.

In this setup, any static scaling results in lower accuracy than solving each task separately (the single-task baseline). The two tasks appear to compete for model capacity, since increase in the accuracy of one task results in decrease in the accuracy of the other. Uncertainty weighting (Kendall et al., 2018) and GradNorm (Chen et al., 2018) find solutions that are slightly better than grid search but distinctly worse than the single-task baseline. In contrast, our method finds a solution that efficiently utilizes the model capacity and yields accuracies that are as good as the single-task solutions. This experiment demonstrates the effectiveness of our method as well as the necessity of treating MTL as multi-objective optimization. Even after a large hyper-parameter search, *any* scaling of tasks does not approach the effectiveness of our method.

## 4.2 Multi-Label Classification

Next, we tackle multi-label classification. Given a set of attributes, multi-label classification calls for deciding whether each attribute holds for the input. We use the CelebA dataset (Liu et al., 2015b), which includes 200K face images annotated with 40 attributes. Each attribute gives rise to a binary classification task and we cast this as a 40-way MTL problem. We use ResNet-18 (He et al., 2016) without the final layer as a shared representation function, and attach a linear layer for each attribute (see the supplement for further details).

We plot the resulting error for each binary classification task as a radar chart in Figure 2. The average over them is listed in Table 1. We skip grid search since it is not feasible over 40 tasks. Although uniform scaling is the norm in the multi-label classification literature, single-task performance is significantly better. Our method outperforms baselines for significant majority of tasks and achieves comparable performance in rest. This experiment also shows that our method remains effective when the number of tasks is high.

Table 1: Mean of error per category of MTL algorithms in multi-label classification on CelebA (Liu et al., 2015b).

|  | Average error |
| --- | --- |
| Single task | 8.77 |
| Uniform scaling | 9.62 |
| Kendall et al. 2018 | 9.53 |
| GradNorm | 8.44 |
| Ours | **8.25** |

## 4.3 Scene Understanding

To evaluate our method in a more realistic setting, we use scene understanding. Given an RGB image, we solve three tasks: semantic segmentation (assigning pixel-level class labels), instance

Table 2: Effect of the MGDA-UB approximation. We report the final accuracies as well as training times for our method with and without the approximation.

| | Scene understanding (3 tasks) | | | | Multi-label (40 tasks) | |
|---|---|---|---|---|---|---|
| | Training time | Segmentation mIoU [%] | Instance error [px] | Disparity error [px] | Training time (hour) | Average error |
| Ours (w/o approx.) | 38.6 | 66.13 | 10.28 | 2.59 | 429.9 | 8.33 |
| Ours | **23.3** | **66.63** | **10.25** | **2.54** | **16.1** | **8.25** |

segmentation (assigning pixel-level instance labels), and monocular depth estimation (estimating continuous disparity per pixel). We follow the experimental procedure of Kendall et al. (2018) and use an encoder-decoder architecture. The encoder is based on ResNet-50 (He et al., 2016) and is shared by all three tasks. The decoders are task-specific and are based on the pyramid pooling module (Zhao et al., 2017) (see the supplement for further implementation details).

Since the output space of instance segmentation is unconstrained (the number of instances is not known in advance), we use a proxy problem as in Kendall et al. (2018). For each pixel, we estimate the location of the center of mass of the instance that encompasses the pixel. These center votes can then be clustered to extract the instances. In our experiments, we directly report the MSE in the proxy task. Figure 4 shows the performance profile for each pair of tasks, although we perform all experiments on all three tasks jointly. The pairwise performance profiles shown in Figure 4 are simply 2D projections of the three-dimensional profile, presented this way for legibility. The results are also listed in Table 4.

MTL outperforms single-task accuracy, indicating that the tasks cooperate and help each other. Our method outperforms all baselines on all tasks.

### 4.4 Role of the Approximation

In order to understand the role of the approximation proposed in Section 3.3, we compare the final performance and training time of our algorithm with and without the presented approximation in Table 2 (runtime measured on a single Titan Xp GPU). For a small number of tasks (3 for scene understanding), training time is reduced by 40%. For the multi-label classification experiment (40 tasks), the presented approximation accelerates learning by a factor of 25.

On the accuracy side, we expect both methods to perform similarly as long as the full-rank assumption is satisfied. As expected, the accuracy of both methods is very similar. Somewhat surprisingly, our approximation results in slightly improved accuracy in all experiments. While counter-intuitive at first, we hypothesize that this is related to the use of SGD in the learning algorithm. Stability analysis in convex optimization suggests that if gradients are computed with an error $\hat{\nabla}_{\boldsymbol{\theta}}\mathcal{L}^t = \nabla_{\boldsymbol{\theta}}\mathcal{L}^t + \mathbf{e}^t$ ($\boldsymbol{\theta}$ corresponds to $\boldsymbol{\theta}^{sh}$ in (3)), as opposed to $\mathbf{Z}$ in the approximate problem in (MGDA-UB), the error in the solution is bounded as $\|\hat{\alpha} - \alpha\|_2 \leq \mathcal{O}(\max_t \|\mathbf{e}^t\|_2)$. Considering the fact that the gradients are computed over the full parameter set (millions of dimensions) for the original problem and over a smaller space for the approximation (batch size times representation which is in the thousands), the dimension of the error vector is significantly higher in the original problem. We expect the $l_2$ norm of such a random vector to depend on the dimension.

In summary, our quantitative analysis of the approximation suggests that (i) the approximation does not cause an accuracy drop and (ii) by solving an equivalent problem in a lower-dimensional space, our method achieves both better computational efficiency and higher stability.

## 5 Conclusion

We described an approach to multi-task learning. Our approach is based on multi-objective optimization. In order to apply multi-objective optimization to MTL, we described an efficient algorithm as well as specific approximations that yielded a deep MTL algorithm with almost no computational overhead. Our experiments indicate that the resulting algorithm is effective for a wide range of multi-task scenarios.

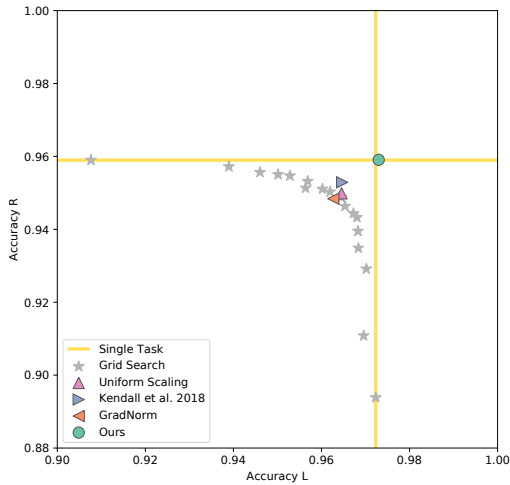

Figure 3: **MultiMNIST accuracy profile.** We plot the obtained accuracy in detecting the left and right digits for all baselines. The grid-search results suggest that the tasks compete for model capacity. Our method is the only one that finds a solution that is as good as training a dedicated model for each task. Top-right is better.

Table 3: Performance of MTL algorithms on MultiMNIST. Single-task baselines solve tasks separately, with dedicated models, but are shown in the same row for clarity.

|  | Left digit accuracy [%] | Right digit accuracy [%] |
|---|---|---|
| Single task | **97.23** | **95.90** |
| Uniform scaling | 96.46 | 94.99 |
| Kendall et al. 2018 | 96.47 | 95.29 |
| GradNorm | 96.27 | 94.84 |
| Ours | **97.26** | **95.90** |

Table 4: Performance of MTL algorithms in joint semantic segmentation, instance segmentation, and depth estimation on Cityscapes. Single-task baselines solve tasks separately but are shown in the same row for clarity.

|  | Segmentation mIoU [%] | Instance error [px] | Disparity error [px] |
|---|---|---|---|
| Single task | 60.68 | 11.34 | 2.78 |
| Uniform scaling | 54.59 | 10.38 | 2.96 |
| Kendall et al. 2018 | 64.21 | 11.54 | 2.65 |
| GradNorm | 64.81 | 11.31 | **2.57** |
| Ours | **66.63** | **10.25** | **2.54** |

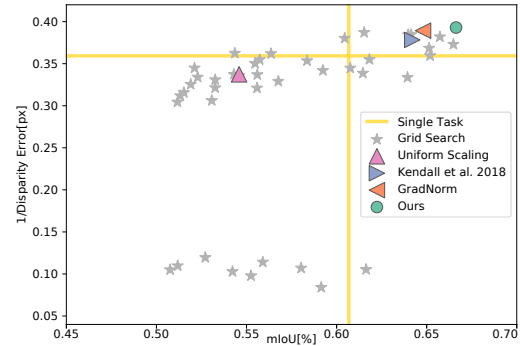

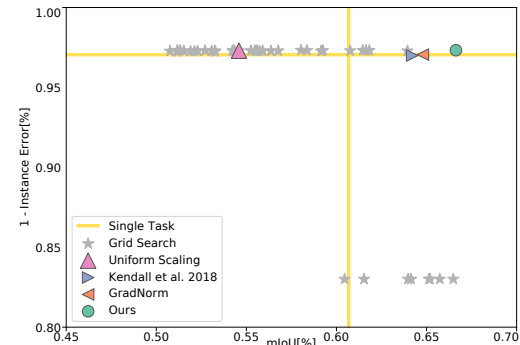

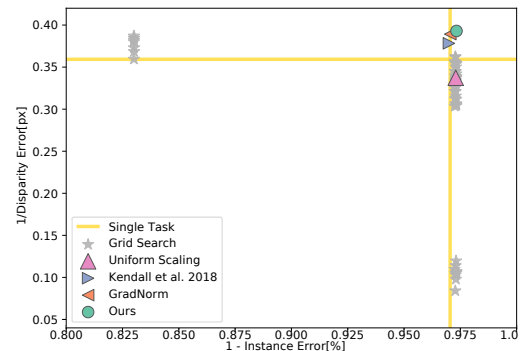

Figure 4: **Cityscapes performance profile.** We plot the performance of all baselines for the tasks of semantic segmentation, instance segmentation, and depth estimation. We use mIoU for semantic segmentation, error of per-pixel regression (normalized to image size) for instance segmentation, and disparity error for depth estimation. To convert errors to performance measures, we use $1 -$ instance error and $1/$disparity error. We plot 2D projections of the performance profile for each pair of tasks. Although we plot pairwise projections for visualization, each point in the plots solves all tasks. Top-right is better.

## Footnotes

[1] This definition can be extended to the partially-labelled case by extending $\mathcal{Y}^t$ with a null label.

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
