[Supplementary Material · appendix_supplementary.pdf]

# A  Proof of Theorem 1

*Proof.* We begin by showing that if the optimum value of (MGDA-UB) is 0, so is the optimum value of (3). This shows the first case of the theorem. Then, we will show the second part.

If the optimum value of (MGDA-UB) is 0,

$$\sum_{t=1}^{T} \alpha^t \nabla_{\boldsymbol{\theta}^{sh}} \hat{\mathcal{L}}^t(\boldsymbol{\theta}^{sh}, \boldsymbol{\theta}^t) = \frac{\partial \mathbf{Z}}{\partial \theta^{sh}} \sum_{t=1}^{T} \alpha^t \nabla_{\mathbf{Z}} \hat{\mathcal{L}}^t = \sum_{t=1}^{T} \alpha^t \nabla_{\theta^{sh}} \hat{\mathcal{L}}^t = 0 \tag{7}$$

Hence $\alpha^1, \ldots, \alpha^T$ is the solution of (3) and the optimal value of (3) is 0. This proves the first case of the theorem. Before we move to the second case, we state a straightforward corollary. Since $\frac{\partial \mathbf{Z}}{\partial \theta^{sh}}$ is full rank, this equivalence is bi-directional. In other words, if $\alpha^1, \ldots, \alpha^T$ is the solution of (3), it is the solution of (MGDA-UB) as well. Hence, both formulations completely agree on Pareto stationarity.

In order to prove the second case, we need to show that the resulting descent direction computed by solving (MGDA-UB) does not increase any of the loss functions. Formally, we need to show that

$$\left( \sum_{t=1}^{T} \alpha^t \nabla_{\theta^{sh}} \hat{\mathcal{L}}^t \right)^{\mathsf{T}} \left( \nabla_{\theta^{sh}} \hat{\mathcal{L}}^{t'} \right) \geq 0 \quad \forall \, t' \in \{1, \ldots, T\} \tag{8}$$

This condition is equivalent to

$$\left( \sum_{t=1}^{T} \alpha^t \nabla_{\mathbf{Z}} \hat{\mathcal{L}}^t \right)^{\mathsf{T}} \mathbf{M} \left( \nabla_{\mathbf{Z}} \hat{\mathcal{L}}^{t'} \right) \geq 0 \quad \forall \, t' \in \{1, \ldots, T\} \tag{9}$$

where $\mathbf{M} = \left( \frac{\partial \mathbf{Z}}{\partial \theta^{sh}} \right)^{\mathsf{T}} \left( \frac{\partial \mathbf{Z}}{\partial \theta^{sh}} \right)$. Since $\mathbf{M}$ is positive definite (following the assumption), this is further equivalent to

$$\left( \sum_{t=1}^{T} \alpha^t \nabla_{\mathbf{Z}} \hat{\mathcal{L}}^t \right)^{\mathsf{T}} \left( \nabla_{\mathbf{Z}} \hat{\mathcal{L}}^{t'} \right) \geq 0 \quad \forall \, t' \in \{1, \ldots, T\} \tag{10}$$

We show that this follows from the optimality conditions for (MGDA-UB). The Lagrangian of (MGDA-UB) is

$$\left( \sum_{t=1}^{T} \alpha^t \nabla_{\mathbf{Z}} \hat{\mathcal{L}}^t \right)^{\mathsf{T}} \left( \sum_{t=1}^{T} \alpha^t \nabla_{\mathbf{Z}} \hat{\mathcal{L}}^t \right) - \lambda \left( \sum_{i} \alpha^i - 1 \right) \quad \text{where } \lambda \geq 0. \tag{11}$$

The KKT condition for this Lagrangian yields the desired result as

$$\left( \sum_{t=1}^{T} \alpha^t \nabla_{\mathbf{Z}} \hat{\mathcal{L}}^t \right)^{\mathsf{T}} \left( \nabla_{\mathbf{Z}} \hat{\mathcal{L}}^t \right) = \frac{\lambda}{2} \geq 0 \tag{12}$$

$\square$

# B  Additional Results on Multi-label Classification

In this section, we present the experimental results we did not include in the main text.

In the main text, we plotted a radar chart of the binary attribute classification errors. However, we did not include the tabulated results due to the space limitations. Here we list the binary classification error of each attribute for each algorithm in Table 5.

Table 5: Multi-label classification error per attribute for all algorithms.

| | Uniform scaling | Single task | Kendall et al. | Grad Norm | Ours | | Uniform scaling | Single task | Kendall et al. | Grad Norm | Ours |
|---|---|---|---|---|---|---|---|---|---|---|---|
| Attr. 0 | 7.11 | 7.16 | 7.18 | 6.54 | **6.17** | Attr. 5 | 4.91 | 4.75 | 4.95 | 4.19 | **4.13** |
| Attr. 1 | 17.30 | **14.38** | 16.77 | 14.80 | 14.87 | Attr. 6 | 20.97 | 14.24 | 15.17 | **14.07** | 14.08 |
| Attr. 2 | 20.99 | 19.25 | 20.56 | 18.97 | **18.35** | Attr. 7 | 18.53 | 17.74 | 18.84 | 17.33 | **17.25** |
| Attr. 3 | 17.82 | 16.79 | 18.45 | 16.47 | **16.06** | Attr. 8 | 10.22 | 8.87 | 10.19 | 8.67 | **8.42** |
| Attr. 4 | 1.25 | 1.20 | 1.17 | 1.13 | **1.08** | Attr. 9 | 5.29 | 5.09 | 5.44 | 4.68 | **4.60** |
| Attr. 10 | 4.14 | 4.02 | 4.33 | 3.77 | **3.60** | Attr. 15 | 0.81 | **0.52** | 0.62 | 0.56 | 0.56 |
| Attr. 11 | 16.22 | 15.34 | 16.64 | 14.73 | **14.56** | Attr. 16 | 4.00 | 3.94 | 3.99 | 3.72 | **3.46** |
| Attr. 12 | 8.42 | 7.68 | 8.85 | **7.23** | 7.41 | Attr. 17 | 2.39 | 2.66 | 2.35 | **2.09** | 2.16 |
| Attr. 13 | 5.17 | 5.15 | 5.26 | 4.75 | **4.52** | Attr. 18 | 8.79 | 9.01 | 8.84 | 8.00 | **7.83** |
| Attr. 14 | 4.14 | 4.13 | 4.17 | 3.73 | **3.54** | Attr. 19 | 13.78 | 12.27 | 13.86 | 11.79 | **11.29** |
| Attr. 20 | 1.61 | 1.61 | 1.58 | **1.42** | 1.43 | Attr. 25 | 27.59 | 24.82 | 26.94 | 24.26 | **23.87** |
| Attr. 21 | 7.18 | **6.20** | 7.73 | 6.91 | 6.26 | Attr. 26 | 3.54 | 3.40 | 3.78 | 3.22 | **3.16** |
| Attr. 22 | 4.38 | 4.14 | 4.08 | 3.88 | **3.81** | Attr. 27 | 26.74 | 22.74 | 26.21 | 23.12 | **22.45** |
| Attr. 23 | 8.32 | 6.57 | 8.80 | 6.54 | **6.47** | Attr. 28 | 6.14 | 5.82 | 6.17 | 5.43 | **5.16** |
| Attr. 24 | 5.01 | 5.38 | 5.12 | 4.63 | **4.23** | Attr. 29 | 5.55 | 5.18 | 5.40 | 5.13 | **4.87** |
| Attr. 30 | 3.29 | 3.79 | 3.24 | **2.94** | 3.03 | Attr. 35 | 1.15 | 1.13 | 1.08 | **0.94** | 1.08 |
| Attr. 31 | 8.05 | 8.40 | 8.40 | 7.21 | **6.92** | Attr. 36 | 7.91 | 7.56 | 8.06 | 7.47 | **7.18** |
| Attr. 32 | 18.21 | 17.25 | 18.15 | **15.93** | 15.93 | Attr. 37 | 13.27 | 11.90 | 13.47 | 11.61 | **11.19** |
| Attr. 33 | 16.53 | 15.55 | 16.19 | 13.93 | **13.80** | Attr. 38 | 3.80 | **3.29** | 4.04 | 3.57 | 3.51 |
| Attr. 34 | 11.12 | 9.76 | 11.46 | 10.17 | **9.73** | Attr. 39 | 13.25 | 13.40 | 13.78 | 12.26 | **11.95** |

Figure 5: Architecture used for MultiMNIST experiments.

## C Implementation Details

### C.1 MultiMNIST

We use the MultiMNIST dataset, which overlays multiple images together (Sabour et al., 2017). For each image, a different one is chosen uniformly in random. One of these images is placed at the top-left and the other at the bottom-right. We show sample MultiMNIST images in Figure 6.

For the MultiMNIST experiments, we use an architecture based on LeNet (LeCun et al., 1998). We use all layers except the final one as a shared encoder. We use the fully-connected layer as a task-specific function for the left and right tasks by simply adding two independent fully-connected layers, each taking the output of the shared encoder as input. As a task-specific loss function, we use the cross-entropy loss with a softmax for both tasks. The architecture is visualized in Figure 5.

The implementation uses PyTorch (Paszke et al., 2017). For all baselines, we searched over the set $LR = \{1e-4, 5e-4, 1e-3, 5e-3, 1e-2, 5e-2\}$ of learning rates and chose the model with the

Figure 6: Sample MultiMNIST images. In each image, one task (task-L) is classifying the digit on the top-left and the second task (task-R) is classifying the digit on the bottom-right.

highest validation accuracy. We used SGD with momentum, halving the learning rate every 30 epochs. We use batch size 256 and train for 100 epochs. We report test accuracy.

## C.2 Multi-label classification

For multi-label classification experiments, we use ResNet-18 (He et al., 2016) without the final layer as a shared representation function. Since there are 40 attributes, we add 40 separate $2048 \times 2$ dimensional fully-connected layers as task-specific functions. The final two-dimensional output is passed through a 2-class softmax to get binary attribute classification probabilities. We use cross-entropy as a task-specific loss. The architecture is visualized in Figure 7.

The implementation uses PyTorch (Paszke et al., 2017). We resize each CelebA image (Liu et al., 2015b) to $64 \times 64 \times 3$. For all experiments, we searched over the set $LR = \{1e-4, 5e-4, 1e-3, 5e-3, 1e-2, 5e-2\}$ of learning rates and chose the model with the highest validation accuracy. We used SGD with momentum, halving the learning rate every 30 epochs. We use batch size 256 and train for 100 epochs. We report attribute-wise binary accuracies on the test set as well as the average accuracy.

Figure 7: Architecture used for multi-label classification experiments.

## C.3 Scene understanding

For scene understanding experiments, we use the Cityscapes dataset (Cordts et al., 2016). We resize all images to resolution $256 \times 512$ for computational efficiency. As a shared representation function (encoder), we use the ResNet-50 architecture (He et al., 2016) in fully-convolutional fashion. We take the ResNet-50 architecture and only use layers prior to average pooling that are fully convolutional. As a decoder, we use the pyramid pooling module (Zhao et al., 2017) and set the output sizes to $256 \times 512 \times 19$ for semantic segmentation (19 classes), $256 \times 512 \times 2$ for instance segmentation (one output channel for the x-offset of the center location and another channel for the y-offset), and $256 \times 512 \times 1$ for monocular depth estimation. For instance segmentation, we use the proxy task of estimating the offset for the center location of the instance that encompasses the pixel. We directly estimate disparity instead of depth and later convert it to depth using the provided camera intrinsics. As a loss function, we use cross-entropy with a softmax for semantic segmentation, and MSE for depth and instance segmentation. We visualize the architecture in Figure 8.

We initialize the encoder with a model pretrained on ImageNet (Deng et al., 2009). We use the implementation of the pyramidal pooling network with bilinear interpolation shared by Zhou et al. (2017a). Ground-truth results for the Cityscapes test set are not publicly available. Therefore, we report numbers on the validation set. As a validation set for hyperparameter search, we randomly choose 275 images from the training set. After the best hyperparameters are chosen, we retrain with the full training set and report the metrics on the Cityscapes validation set, which our algorithm never sees during training or hyperparameter search. As metrics, we use mean intersection over union (mIoU) for semantic segmentation, MSE for instance segmentation, and MSE for disparities (depth estimation). We directly report the metric in the proxy task for instance segmentation instead of performing a further clustering operation. For all experiments, we searched over the set $LR = \{1e-4, 5e-4, 1e-3, 5e-3, 1e-2, 5e-2\}$ of learning rates and chose the model with the highest validation accuracy. We used SGD with momentum, halving the learning rate every 30 epochs. We use batch size 8 and train for 250 epochs.

Figure 8: Architecture used for scene understanding experiments.