[Reviews · NeurIPS 2018]

Reviewer 1



This work is interesting. I would like to ask the authors to compare their approach to GradNorm. Comparison to Kendall et. al is not sufficient due to caveats of uncertainty weighting. Also, the authors are strongly encourage on NYC V2 data set. This works lacks a stronger experimental section. I have re-read the paper and the rebuttal posted by the authors. I thank them for performing the additional experiments and comparison they did with GradNorm. They did, however, ignore to do demonstrate the performance of their approach on NYU V2 dataset. Furthermore, here is a more detailed reviewed i chose to omit initially. A large portion of the introduction in spent on discussing why assuming a linear combined loss function L = sum(w_iL_i) is not ideal for multitask learning, yet their method is choosing this linear combination of gradients based on their numerical \alpha vector. My high-level opinion is: I think it's a neat idea that makes a valiant effort to take a principle approach on MTL. The results are also promising, The main contribution however, is the approximation of the single backward pass, which was the chain rule plus an application of the triangle inequality. A few more specific comments: (1) I believ from the Desideri paper, that there is a 1d search for the learning rate at each step - they set learning rate to the maximum learning rate possible such that all task losses are monotonically decreasing at that learning rate. In this work, however, the 1d search is not present, despite the paper borrowing heavily from this paper. Is there a justification for this? (2) How does the algorithm interact with more complex update rules? E.G. Adam, momentum? (3) I am still confused by the approximation performing better than the non-approximated version. How is the approximation a form of regularization?

Reviewer 2



Overall summary of the paper: This paper proposed a multi-task learning algorithm from multi-objective optimization perspective and the authors provided an approximation algorithm, which could accelerate the training process. The authors claim that existing MTL algorithms used linear combinations (uniform weight) of the loss from different tasks, which is hard to achieve the Pareto optimality. Unlike the uniform weight strategy, the authors use the MGDA algorithm to solve the optimal weight, which would increase the performance for all the tasks to achieve the Pareto optimality. Moreover, when solving the sub-problem for shared parameters, the author gave an upper bound of the loss function, and this upper bound optimization only requires one-time back propagation regardless of the number of tasks. The results show that the approximation strategy not only can accelerate the training process but also improve the performance. Strengths: 1. The proposed method can solve the optimal weight for the loss of different tasks. 2. The approximation strategy can efficiently accelerate the training process. 3. The proposed algorithm is easy to implement. Weaknesses: 1. The motivation is not convincing enough. The decrease of training loss does not mean the generalization performance is also good. Pareto optimality is for the training loss decrease, which has no guarantee that it will also improve the generalization performance. 2. The optimization treatment is not novel. 3. The approximation provides a good and easy way to accelerate the algorithm, which is the spotlight of this paper. However, in the experiment, the approximation one is even better than the original one in prediction, which does not make sense. Although the authors provide some qualitative reasons at the end of the paper, it is hard to convince me why this approximation can even improve the prediction result. I noticed that in Table 2, the training time for without approximation is longer than 10 days. Is the baseline without approximation carefully tuned? 4. The author does not include the uniform scaling + approximation strategy. I am wondering how this baseline perform compared to other methods in the paper. This could help readers to understand which part is really effective.

Reviewer 3



Main idea: This paper investigates the problem of multi-task learning using the multi-objective optimization. The objective is a vector-valued loss that is also applicable to use KKT condition to solve. The authors propose to approximate this constraint convex problem by introducing a shared representation function. Strengths: • The writing is straightforward to follow and the motivation, as well as the related work, are clear. • The geometry visualizations regarding the optimization objective of two-task learning are helpful for the understanding of the general audience. • The notations and formulations are neat and corresponding explanations are proper. • The experiment shows several promising results and the experimental settings are clear and the results are convincing. Weakness: • The proposed framework is the incremental work that ensembles the multi-objective learning and constrained convex optimization and fit it into the multi-task learning scenario. The novelty only comes from the problem approximation which only uses very limited space to elaborate the contributions. • Why the multiple tasks share the same set of examples as the notations state in section 3.1. The different learning tasks could have their own sets of examples and the number of examples could be also very different. Wouldn’t it be the general learning settings of multi-task learning? • How good is the approximation that only has one backward pass in terms of the theoretical analysis? It would be a strong plus if you prove its effectiveness and efficiency in theory. • The experiments look solid and but lack the wide range of comprehensive comparison. Several import baselines are missing. Please mention and compare with the important methods implemented in MALSAR [1] to claim the effectiveness of the proposed framework. Quality: Overall, the writing quality is good, the main idea is clearly presented. Clarity: The notations are well-organized and easy to interpret. Originality: It is an incremental work with limited novelty. Significance: Theoretical proofs are missing, experiments should add more powerful baselines. [1]. J. Zhou. MALSAR: Multi-task learning via structural regularization.